# Feral Kinetics and Cattle Research Within Planetary Boundaries

**DOI:** 10.3390/ani13050802

**Published:** 2023-02-23

**Authors:** Nathalia Brichet, Signe Brieghel, Frida Hastrup

**Affiliations:** 1Department of Veterinary and Animals Sciences, University of Copenhagen, 1870 Frederiksberg, Denmark; 2The Saxo Institute, University of Copenhagen, 2300 Copenhagen, Denmark

**Keywords:** critical kinetics, planetary boundaries, technical solutions, feed additives, livestock, Jevons’ paradox, Denmark, anthropology

## Abstract

**Simple Summary:**

This commentary advocates a note of caution with regard to using the manipulation of feed to address and solve cattle’s negative impacts on the climate and environment. It identifies some of the potential consequences—intentional and unintentional—of this type of ‘solutionism’ and proposes a wider discussion about reducing livestock numbers and thinking with planetary boundaries to promote sustainable animal production systems. The argument is illustrated through findings from an interdisciplinary research project on cattle production in Denmark.

**Abstract:**

The increased attention drawn to the negative environmental impact of the cattle industry has fostered a host of market- and research-driven initiatives among relevant actors. While the identification of some of the most problematic environmental impacts of cattle is seemingly more or less unanimous, solutions are complex and might even point in opposite directions. Whereas one set of solutions seeks to further optimize sustainability pr. unit produced, e.g., by exploring and altering the relations between elements kinetically moving one another inside the cow’s rumen, this opinion points to different paths. While acknowledging the importance of possible technological interventions to optimize what occurs inside the rumen, we suggest that broader visions of the potential negative outcomes of further optimization are also needed. Accordingly, we raise two concerns regarding a focus on solving emissions through feedstuff development. First, we are concerned about whether the development of feed additives overshadows discussions about downscaling and, second, whether a narrow focus on reducing enteric gasses brackets other relations between cattle and landscapes. Our hesitations are rooted in a Danish context, where the agricultural sector—mainly a large-scale technologically driven livestock production—contributes significantly to the total emission of CO_2_ equivalents.

## 1. A Commentary on Feed Additives as Solutions to Cattle’s Climate Impact

Since 2006, when the much-quoted FAO report Livestock’s Long Shadow [1] was published, increasing attention has been paid to the environmental challenges brought about by the cattle industry [2,3]. Problems pile up. Today, agriculture occupies about 38% of Earth’s terrestrial surface [4]. Industrial livestock requires enormous amounts of feed, the production of which takes up more than three quarters of all agricultural land on the planet [5]. This leads to massive deforestation, which proliferates into yet other problems such as soil erosion, nutrient leaching, and loss of biodiversity. Further, the gases emitted from the rumination processes of cattle make up a substantial portion of the greenhouse gases (GHG) that spread through the atmosphere, heating up our globe. In particular, enteric fermentation has been targeted and understood as key to controlling and potentially reducing the climate impact of cattle. In response, a host of market- and research-driven initiatives among relevant actors has emerged. As this special issue suggests, the time has now come to collect knowledge on forage and feedstuff digestion kinetics in ruminants in order to meet the mounting pressures to reduce enteric methane production. 

The fact that domesticated animals produced around the world (mainly cattle, pigs, and poultry) now outnumber wild mammals and birds by a factor of ten no doubt adds pressure to this predicament [5]. Yet, for all the weight this ratio puts on the world’s ecosystems, the scientific knowledge base supporting the industry’s transition towards more sustainable futures most often comes from very specific areas of the natural sciences. Nourishing this quantitatively large global production of a few domesticated animal species, then, is the continuous production of scientific knowledge and research related to issues such as feed intensification, genetics, and health. For cattle, the obvious aim is to further optimize and increase dairy and beef production—industries that in Euro-American production systems are premised on principles of high cost efficiency, achieved by keeping large animal herds on small but intensively managed areas, with easy access to feed products that are often both nutritionally optimized and imported from overseas. 

While the identification of some of the most problematic environmental impacts of cattle is seemingly more or less unanimous, solutions are complex and might even point in opposite directions. As is clear from this special issue, one set of solutions points to interventions into the causal relations of elements kinetically moving one another inside the cow’s rumen. More specifically, feed additives of various sorts are developed to decrease the generation of methane, thus targeting climate change. Surely, these interventions are valuable—after all, in these critical, increasingly hot times, why oppose GHG reductions in whatever form and by whatever means? Nonetheless, in this short contribution, we raise two concerns regarding a focus on solving emissions through feedstuff development. Whereas knowledge about the possibilities of technological optimization is important, we suggest that broader visions of the possible negative outcomes of further optimization are also needed. Notably, we are concerned for two reasons: First, the development of feed additives may overshadow discussions about downscaling. Second, a narrow focus on reducing enteric gasses by manipulating a set of kinetic causal processes inside the rumen may bracket other relations between cattle and landscapes. Both of these consequences, we suggest, can potentially limit the scope of climate impact mitigation in relation to cattle. Below, we substantiate these reservations and go on to probe how the development of feed additives sit with ideas about absolute sustainability and safe operating spaces for humanity. Our approach to the issue of feed additives and cattle production is rooted in anthropology and cultural history. We thus work using a method of ethnographic fieldwork among stakeholders engaged in Danish cattle production, and we conduct document analysis of, e.g., policies and public discussions, just as we look to archival and historical literature to trace earlier connections between state making and livestock production in Denmark.

## 2. Intensified Danish Livestock Production—With Feral Effects

Within the agricultural sciences, an answer to the problematic climate impact of cattle has often been to further intensify animal production cf. [6]. This response, however, rests on a modernistic assumption that more-than-human lives are essentially controllable by humans. The ecological crisis that we are presently witnessing does indeed testify to human activities and projects on a massive scale, but not to human control over causal relations in the more-than-human world. This is what we mean by *feral* kinetics in this opinion’s title; we want to indicate that projects in which (causal) relations were once set in motion by humans through intentional projects often spur various unintended effects [7]. Surely, no one set out to change the climate through animals. A brief historical look at the Danish context in which our research is rooted shows how intensification has come about gradually and in response to a host of societal, political, and historical circumstances that jointly make up what livestock came to be. In Denmark, the agricultural sector is responsible for 27.1% of national GHG emissions (excl. land use, land use change, and forestry (LULUCF)), of which intensive livestock production is a prominent contributor [8]. Danish livestock production is often positively framed in public discourse as economically and environmentally cost-efficient when measured at the scale of singular products. Yet, there is reason to question the hidden costs of this mode of calculation, which often fails to include GHG emissions and global environmental impact derived from the livestock industry’s dependency on feedstuffs, and thus, on land cultivation and deforestation both in Denmark and overseas, as scientists have also pointed out [9]. In Denmark, as in other European countries, agricultural production has historically informed the organization of government, business, and civil society, and politicians still identify Denmark as “a farming country”—see, for example, [10]. It was through the workings and on-going development of the agricultural sector that Denmark as a nation underwent some of its most significant historical changes, not least that of industrial modernization [11,12]. More specifically, 19th- and 20th-century industrial modernization in the agricultural sector came about through the emergence of a strong, export-oriented livestock industry—see, for example, [13]. For instance, the number of pigs in Denmark has increased from 301,000 in 1860 to 12.2 million today, which is now more than twice the Danish population [14,15]. The historical emergence of a strong livestock sector not only changed the welfare and fortuity of rural communities. It also changed the physical appearance of the animal bodies that would increase the reach of Danish industrial interests by yielding more meat and milk per animal than they ever had before, which was achieved, in particular, through the industry’s adoption of scientific approaches to rational and optimized feeding [16,17]. Our point is that during the same historical processes that have made Danish agriculture synonymous with a large livestock industry, the livestock industry has itself become synonymous with high export and feedstuff dependency, as both are integral to how cost efficiency is construed in the industry. However, if the purpose of novel feed additives is to mitigate the climate impact of cattle production, it seems pivotal to ask whether such a high dependency on feeds should itself become a site of scientific intervention in the industry.

## 3. The Issue of Scale and Re-Bound Effects

Indeed, to pinpoint our first reservation, focusing on greening the enteric digestion process by making ‘enhanced’ rumens less harmful, and politically prioritizing this effort, may engender a so-called rebound effect where decreased methane emission pr. cow is evened out by an increased scale of production—a mechanism sometimes summarized as Jevons’ paradox. In other words, addressing the issue of methane emission through conceptualizing the rumen as a discrete and singular site of intervention may occlude the concern both with a potential rise in scale and with other the well-known, environmentally harmful effects of maintaining present-scale livestock production [18]. Inadvertently, we fear, the development of feed additives may contribute to the status quo, thus foreclosing difficult discussions about how we might best use arable land (for feed or food), and about the number of cattle the world’s ecologies (including atmospheric greenhouse gasses) can actually support. Aspiring for sustainability—as genuine change—on a planet with limited resources and imminent tipping points requires that we think in absolute terms. Adjustment and improvement miss the mark.

## 4. The Issue of Other Ecological Relations

Our second hesitation is that a narrow focus on kinetics inside the rumen may eclipse wider ecological relations put into motion by the altered processes in the rumen. As the many ecological crises we are witnessing make clear, humans can no longer be seen as being in full control of various processes on earth—what we referred to above as the ‘feral’ nature of the non-human world in this day and age. Processes once perceived as controllable have proven not to be so, resulting in unintentional global effects that are both distributed and caused by human actions [7]. With regard to feed additives specifically, these may work on other relations than those within the rumen, making it urgent to widen the scope of research. We must be careful not to make yet another potent feral product by limiting our view of what the problems and solutions are.

## 5. Thinking with Absolute Sustainability

To summarize, our hesitations with regard to feed additives concern, first, how additives may maintain status quo in terms of the scale of cattle production, and second, how they may engender unanticipated ecological effects. On both accounts, we suggest, there is a need to consider how technological and natural scientific solutions to the methane issue relate to a planet with boundaries that limit a safe operating space for humanity [19,20]. In short, we want to question how feed additives as a means of reducing methane emissions sit with ideas about absolute sustainability [21,22]. 

If, as we suggest, technological solutions ‘work back’ on the identification of what the problem even is, the result is too easily a circular argument; we can only see the problems that we think we can solve by way of technology—rather than by a just and green societal transition, sanctioned in progressive politics. In making this argument, and exploring how it works in a Danish context, we draw on the model of planetary boundaries originally suggested by Rockström et al. [20]. In 2017, Campbell et al. [18] further worked with the model, arguing that agricultural production is, indeed, a main driver for the eco-system changes occurring, particularly within spheres where the planetary boundaries are transgressed. Our point is simply this: If the impact of agricultural production already exceeds ‘permissible’ limits, something has to change fundamentally. Making things relatively better is just not enough. This is particularly important in agro-industrially intensified countries such as Denmark where livestock industries are (dis)proportionately large; we must question both issues of scale and of unintended side effects. Choices remain to be made that observe planetary boundaries. Now, the scientists and companies who develop feed additives would probably agree that this is just one tool among many others. We want to stress that it is not feed additive development per se that we take issue with. Rather, looking to the recent political work of pushing for a green transition of the Danish cattle industry, we are concerned with the way feed additives have emerged on the political stage. Here, additives are presented as the obvious solution to a universal problem. However, the problem that additives are key to solving—i.e. a massive cattle industry, nourished by feed that causes deforestation and in a monocultural logic—can remain untouched. Further, the wider effects of the additives have yet to be documented.

## 6. A Couple of Examples

To substantiate our argument, below we will look at a couple of instances where feed additives are discussed in a Danish context. We do so via ethnographic fieldwork, where we engage with stakeholders such as researchers and politicians, as well as with various written sources. Our method is ethnographic in the sense that we trace relations and generate analyses in dialogue with the field; as such, we explore what feed additives may be as they are produced and discussed by people who develop, implement, or entrust them with positive effects for mitigating climate change—see also [23]. In 2021, all political parties but one represented in the Danish parliament signed an agreement on the green transition of agricultural production in Denmark [24]. The agreement, launched as historic and ambitious, commits to reducing GHG emissions from agriculture by 55–65% by 2030 compared to 1990 levels, in addition to reducing nitrogen and phosphorous run-off into waterways in order to comply with EU regulation. To reach this binding target, curiously, the agreement starts with listing a number of caveats, all to the effect that the goal must be achieved without decreasing agricultural productivity, nor compromising public finances and Denmark’s competitive edge with regard to agriculture. Instead, the agreement highlights the continued prioritization of developing and implementing new technologies. Indeed, trust in (near) future technological solutions is so great that the historically ambitious deal, as it is now, ensures less than a third of the promised reduction. More precisely, the agreement specifies a reduction of 1.9 tons of CO_2_e out of the 6.1–8 tons which is the overall target for the agricultural sector, equaling a reduction in GHG emissions of 55–65% from 1990 emission levels. The rest of the required reduction, the agreement states, will be brought about by new technologies that have yet to be developed and implemented. Two overall arenas for technological innovation are singled out in the political agreement: namely, the curbing of emissions from manure from all production animals and enteric fermentation in livestock. The agreement states as follows: ”It will be a continued priority that new tools, such as feed additives, are transferred as quickly as possible to the implementation track, and that the demand [for reduced GHG emissions] is adjusted according to what can be realized” [24] (p. 4). 

What we want to point to here is that the means for a very large proportion of the CO_2_ reduction that the agreement commits to have yet to be invented, and that reduction targets are adjustable. In other words, the binding and historic agreement on the green transition of Danish agriculture is highly negotiable, and, further, dependent on uncertain technologies. All the while, the agreement lists a number of other priorities that are not up for negotiation—such as productivity, employment, public finances, and rural development. This leaves it up to innovative technologies to find the remaining (majority) of the promised GHG reduction. Accordingly, as we see it, there is a substantial risk that the agreement’s limited focus on climate change mitigation will lead to so-called “burden-shifting”—see [22]—as the negative impacts of feed production and consumption are only considered with regard to a single planetary boundary, as opposed to asking how feeds can become sustainable in an absolute sense, heeding all biospheres. This is to say that optimizing feeds as a means to mitigate GHG emissions specifically risks overlooking equally important and environmentally detrimental processes such as the eutrophication or acidification of waterbodies. If global feed production and consumption on the whole are largely left unchanged, the use of feed additives to mitigate GHG emissions risks relocating instead of actually solving the problems caused by livestock production. By leaving it up to hopeful investments in future technologies to reduce GHG, we are not forced to consider the number of cows, nor the other effects of an unchanged scale.

Another example from our fieldwork sheds light on the potential unintended effects of implementing feed additive solutions. Below, we provide more detail regarding some of the ways that feed additives work in the practices and discussions of industrial agriculture stakeholders. At the annual Cattle Congress 2022, the head of the Cattle Section in SEGES Innovation—the Danish agricultural interest organization’s independent research unit—together with a researcher from the same organization, gave a talk under the headline “Climate Requirements in the Agricultural Agreement”—the same deal mentioned above. From her point of view, climate requirements can be answered by two distinct means of action: the handling of digestion and manure—while not compromising another distinct theme, namely animal welfare. In this way, she framed the problems involved in having an agricultural sector accounting for over one fourth of Danish GHG emissions by setting a very particular triangular frame within which one should think, talk, research, and act in relation to climate requirements: welfare, digestion, and manure. 

She mentioned that feeding with additives could reduce 20% of the 0.17 million tons CO_2_ that needed to be reduced by 2025, but also expressed frustration with the limited amount of money reserved to introduce and implement a new product approved by the European Union in the spring of 2022. New routines on farms need to be developed and supported, and potentially skeptical farmers should be convinced that milk yields will not decrease on account of the new additive. The researcher assisted her and elaborated on the tools needed to reach the goals in the climate law and the agreement discussed above. These were tools that altogether confirmed the dictum of ‘more for less’ (higher yield and more efficiency in feed and in producing bodies) that has made Danish livestock production competitive on a global market despite high production costs. Just as important, the presentation repeated a dictum that has recently become a standard answer to green goals: optimization equals sustainability. The researcher then went on to talk about the possibilities of reducing methane by changing diets—rapeseed and a handful of feed additives were mentioned, along with the possibilities and challenges these feeding options spurred. He singled out one new product and stated that the climate impact of milk will decrease by 17%, adding that if all conventional producers would implement the additive, the reduction targets for 2030 could be met. Thus far, he continued, the additive has only been tested on Holstein cattle. He wrapped up his presentation by saying that if any of the farmers present were interested in testing the product on their animals, they should feel free to get in touch. 

What interests us here is that the talk can be seen as combining the launch of a solution with calling for further tests, thereby mimicking the agreement above in its expectations for future effects of something yet to be fully developed and tested. Interestingly, a person in the audience raised his hand and questioned the manure from cattle fed with the approved additive, asking whether the emissions from it altered when distributed on the fields. In response, the researcher answered that the amount of product used is so small, and further that the product is processed so quickly in the cow that it was very unlikely that it would have an effect elsewhere, outside the rumen. However, he continued, researchers in Canada have recently conducted a study where they pointed to higher emissions from manure in the fields as an effect of the application of feed additives. Interestingly, this Canadian study—or hesitation, we could call it—did not seem to ‘alter’ the Danish researcher’s hopes for feed additives once they have been further tested. Chatting with the researcher after the talk, it became clear that potentially increased emissions on the fields were understood as a problem for another research field—namely, that which deals with manure handling. For him, it seemed, there were so many kinetic relations to be explored within the rumen, and understanding what happens later, out on the fields, would be a theme to be researched once processes in the rumen are more fully understood. Our point here is to highlight the decoupling of what goes on in the rumen upon applying feed additives from the ‘afterlife’ of such an intervention. 

## 7. Conclusion: Heeding all Biosphere Domains at Once

To conclude, what we argue—and the reason for our reservations towards the prevalent kind of solutionism offered by the development of feed additives—is that it takes a very particular perspective on the cow for its rumen to be the sole target of any intervention. From this perspective, the cow is a singular unit from within which technology can decrease methane emission. To the contrary, as anthropologists, we would see any cow as a set of relations, ranging from the microbial level to global issues of deforestation [23]. While we do not oppose feed additives as such, we do hold that they risk building on and maintaining a tunnel vision, as also described above, that allows for a curious disconnection of cattle’s rumen from other cattle-related processes and decisions, including the discussion of scale and other effects than GHG emissions. In Denmark, and elsewhere, other biosphere problems are also apparent as seen from the model of planetary boundaries. Not least, we have huge problems with the leaching of N and P severely affected waterways in Denmark, as a result of the scale of animal production, regardless of its climate efficiency when measured pr. kilogram of, e.g., milk. Put bluntly, as we see it, if we care about the immediate threats to the safe operating space for humanity, it makes little sense to assess the climate impact of cattle pr. singular rumen. One direct insight from the principle of absolute sustainability is that all agricultural resource activities impact many of the nine biosphere domains. Accordingly, solutions need to follow suit. We cannot afford the luxury of solving one problem at a time. 

## Data Availability

Not applicable.

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
