# Peer review of "Feral Kinetics and Cattle Research Within Planetary Boundaries"

_animals, 2023, doi:10.3390/ani13050802_

Round 1

Reviewer 1 Report

This short commentary advocates caution about the promise of manipulating livestock feed to address environmental harms. It identifies some of the potential consequences of this type of ‘solutionism' – both intentional and unintentional – and advocates for a wider conversation about reducing livestock numbers. The argument is illustrated with reference to developments in the Danish livestock industry. This commentary has potential and would offer a valuable alternative perspective to this special edition. It should be published, but would be enhanced by attending to the following concerns:

1.     The paper needs to define and explain what is meant by feral at the start, given the significance of the term in the title. On page 3 it should illustrate what is meant by ‘going feral’ and ‘feral products’ in the specific case of livestock agriculture.

2.     The argument and the structure of the paper are unclear. There is rather too much signposting at the start of what will follow for such a short article. This leads to a fair amount of repetition. The paper tells us what it is going to do, then it does it all over again. This repetition could be ironed out.

3.     Even as someone knowledgeable about this field, I feel the paper assumes too much prior knowledge of the role of livestock in various emissions profiles, as well as of the Danish context. A general, short introduction would be helpful here.

4.     The authors could provide a little more introduction to their positionality, as well as a short background on the research project and methodology that underpins this intervention.

5.     The paper could make more use of existing literatures, including work on:

a.     the deficiency of solutions focused approaches, e.g. Clay, N., A. E. Sexton, T. Garnett and J. Lorimer (2020). "Palatable disruption: the politics of plant milk." Agriculture and Human Values 37(4): 945-962; Goldstein, J. (2018). Planetary Improvement: Cleantech Entrepreneurship and the Contradictions of Green Capitalism. Cambridge, Mass., MIT Press.

b.     wider work on livestock futures. E.g. McGregor, A. and D. Houston (2018). "Cattle in the Anthropocene: Four propositions." Transactions of the Institute of British Geographers 43(1): 3-16; Garnett, T., C. Godde, A. Muller, E. Röös, P. Smith, I. d. Boer, E. z. Ermgassen, M. Herrero, C. v. Middelaar, Christian Schader and H. v. Zanten (2017). Grazed and confused? Ruminating on cattle, grazing systems, methane, nitrous oxide, the soil carbon sequestration question – and what it all means for greenhouse gas emissions FCRN. www.oxfordmartin.ox.ac.uk/downloads/reports/fcrn_gnc_report.pdf

c.     Including work on microbial manipulation in the rumen, e.g. Cooper, M. H. (2017). "Open Up and Say “Baa”: Examining the Stomachs of Ruminant Livestock and the Real Subsumption of Nature." Society & Natural Resources 30(7): 812-828; Folkers, A. and S. Opitz (2022). "Low-carbon cows: From microbial metabolism to the symbiotic planet." Social Studies of Science 52(3): 330-352.

Minor points

Pg 2 ln 83 explain the acronym LULUCF

Pg 4, line 153 ‘ensures only a reduction of 1,9 ton CO2e out of the 6,1-8 tons 153 which is the overall target for the agricultural sector equaling the decision to reduce GHG 154 by 55-65% of 1990 emission levels.’ Not clear what is being referred to here in terms of quantity and time period? This figure seems quite low. This claim also needs some reference.

P4 line 176. Who are SEGES Innovation?

P4 line 181 ‘an agricultural sector emitting over 1⁄4 of the Danish inventory’. Inventory of what?

P4 line 198 ‘skeptic’ should be sceptical.

Author Response

Reviewer 1

Thank you very much for these very helpful comments. They have surely improved the Commentary. In addition to mending the minor poitns, we have addressed the more comprehensive suggestions as follows:

  1. We have explained what is meant by ’feral’ and why we have it as part of the title.
  2. This was an important point. We have revised the structure of the paper, so that the flow is now better, and repetitions have been weeded out – there is now a clearer progression between secetions. We have inserted sub-headings to make it easier to follow the line of thought.
  3. Good point. We have inserted a general introduction to the Danish context, backed by references for further reading. Hopefully, this makes it easier to appreciate the empirical examples.
  4. We have added a brief description of our disciplinary backgrounds and methods to make our approach and positionality more apparent.
  5. The suggested literature was very relevant and useful, and we have made some use of it here. Given that the piece is a rather short commentary, we have to save a fuller processing the readings for some other occasion.

We much appreciate the comments – thank you!

Reviewer 2 Report

I find it very interesting and important that the principles of absolute sustainability is used outside technical fields in this commentary. You wrap very well up (what I think) is the most important feature with the absolute sustainability assessment tool; that we can assess the sustainability of a product/system in absolute terms, so we do not go on and on optimizing a process that will never become sustainable.

You also address that we should not only limit the sustianability discussion to climate change impacts, but other environmental impacts should also be considered. This is also an important feature of the absolute sustainability tool. You could also highlight this feature in the discussion and add that it aims to prevent “burden-shifting”. I.e. that optimizing only considering GHG emissions, could risk that we oversee associated increase in other impacts such as eutrophication or acidification. For example, if the additives may lead to an increase in emissions of other gasses than GHGs (my knowledge of additives and their effect is limited, so you might have a better example). Finally, you address the problem that the fields are operating separately, and that the potential effects of feed additives outside the rumen (e.g. when applying the manure on the field) is considered another fields problem. This is also an important point, and something absolute sustainability assessments aims to tackle, by considering the entire life cycle of the product/process in question. This could also be highlighted in the part where “tunnel vision” and fields working separately is discussed.

Other minor comments:

-          Simple summary and abstract are identical.

-          Line 44; “ Problems crowd”? I don’t understand this.

-          “While the identification of some of the most problematic environmental impacts of cattle is seemingly more or less unanimous,” I miss some explanation what the most problematic environmental impacts actually are.

-          In general, write “CO2” as “CO2

-          Line 83: please write out the abbreviation “LULUCF” when it is used the first time.

-          Line 92: Maybe you could add an example (and citation) of “rebound-effect” from other products or industries

-          Line 125: State which planetary boundaries are transgressed?

-          I think some more explanation of absolute versus relative sustainability is needed, for example after you use the term “absolute” the first time in line 100. Also the Planetary Boundary framework should be explained a bit more.

Author Response

Reviewer 2

Thank you very much for the positive review. The piece is clearly read as it was intended, which is in itself helpful. We have attended to the small points raised, in addition to giving the paper as a whole a careful revision, so hopefully the minor points are seen to.

Reviewer 3 Report

Hello , 

This Commentary paper :Feral Kinetics and Cattle Research within Planetary Boundaries, rise very important issue of effect of industrial production of farm animals on environment. The case is focus heavily on Danish reality but it can be very easy related to global problem. The paper rise important questions. And we should consider them in our decision for future solution on mitigation the effect of animal production on environment.    

Author Response

Reviewer 3

Thank you very much for this positive review. It is a great help to know that the commentary is read as it is intended.